# Association between Different Grades of Myopic Tractional Maculopathy and OCT-Based Macular Scleral Deformation

**DOI:** 10.3390/jcm11061599

**Published:** 2022-03-14

**Authors:** Jingyang Feng, Ruonan Wang, Jiayi Yu, Qiuying Chen, Jiangnan He, Hao Zhou, Yuchen Du, Chen Liu, Weijun Wang, Xun Xu, Xian Xu, Ying Fan

**Affiliations:** 1Department of Ophthalmology, Shanghai General Hospital, Shanghai Jiao Tong University, Shanghai 200080, China; jingyang.feng@shgh.cn (J.F.); lww@sjtu.edu.cn (R.W.); jiayi_nicole@sjtu.edu.cn (J.Y.); izzie_qiu@sjtu.edu.cn (Q.C.); haomayday@163.com (H.Z.); liuchen_1027@sjtu.edu.cn (C.L.); weijun.wang@shgh.cn (W.W.); drxuxun@sjtu.edu.cn (X.X.); 2National Clinical Research Center for Eye Diseases, Shanghai 200080, China; 3Shanghai Key Laboratory of Ocular Fundus Diseases, Shanghai 200080, China; 4Shanghai Engineering Center for Visual Science and Photomedicine, Shanghai 200080, China; 5Shanghai Eye Disease Prevention and Treatment Center, Shanghai 200080, China; hejiangnan85@126.com; 6School of Electronic Information and Electrical Engineering, Shanghai Jiao Tong University, Shanghai 200240, China; yuchendu@sjtu.edu.cn

**Keywords:** high myopia, OCT, myopic tractional maculopathy, atrophic myopic maculopathy, macular outward scleral height

## Abstract

Purpose: To investigate the characteristics of macular outward scleral height (MOSH) in different grades of myopic tractional maculopathy (MTM) and explore the risk factors for MTM. Methods: A total of 188 eyes (188 participants) with high myopia were divided into the no MTM (nMTM) group and the MTM group, which was further graded into foveoschisis, foveal detachment, full-thickness macular hole, and macular hole with retinal detachment. Swept-source optical coherence tomography was used to measure the MOSH. Results: No significant differences were found in axial length between the nMTM and MTM groups (*p* = 0.295). The MOSH was significantly higher in the MTM group (*p* < 0.001), which was identified as a risk factor for MTM (OR = 1.108, *p* < 0.001). The proportion of eyes with severe atrophic myopic maculopathy (AMM) was higher in the MTM group (28.48%) (*p* = 0.003). The macular hole with foveoschisis (MH/FS+) subgroup presented a higher average MOSH (*p* = 0.012) and more severe AMM (*p* = 0.009) than the macular hole without foveoschisis (MH/FS−) subgroup. Conclusion: MOSH would be more suitable for estimating MTM occurrence than axial length. The grading of AMM helps to evaluate the severity of MTM. The categorization of MH/FS− as a distinct grade from MH/FS+ might be preferable.

## 1. Introduction

Myopic tractional maculopathy (MTM), first reported by Panozzo et al. in 2004, is a primary complication that causes blindness in patients with high myopia. MTM represents a spectrum of disorders related to vitreomacular traction (VMT) [1]. In 2018, Ruiz-Medrano et al. introduced a novel ATN grading system, which contained three important components of myopic maculopathy, including atrophy (A), traction (T), and neovascularization (N) [2]. The T component mainly includes a lack of macular schisis, foveoschisis (FS), foveal detachment (FD), full-thickness macular hole (MH), and MH with retinal detachment (MHRD). MTM progression begins with the slow development of stable FS, wherein vision is unaffected, or only slightly impaired. When MTM progresses to FD, or becomes more severe, lesions begin to develop rapidly. Irreversible visual impairment may occur if a timely clinical intervention is not administered [3,4]. Therefore, investigating the risk factors for MTM, as well as the physiological characteristics, especially in the development from mild to severe stages, in eyes with high myopia is essential.

Posterior staphyloma (PS) is considered a hallmark of pathologic myopia and an important cause of myopic maculopathy. Previous studies have reported a high incidence of MTM in myopic eyes with PS, due to the traction from the deformation of macular sclera [5,6]. However, whether the exacerbation from FS to MHRD in MTM is correlated with the morphologic change in macular sclera needs further exploration. Recently, the posterior staphyloma height (PSH), which is defined as the distance from the RPE line beneath the fovea to the edge of the OCT scan, has been proposed as a means to objectively quantify the severity of scleral curvature in the macular [7]. The confidence and feasibility of OCT-based PSH measurements have been confirmed [8].

When depicting the scleral deformation in the macular in this study, we found that the term PSH is not accurate enough, due to the following reasons: (1) the PS may not center on the macula; (2) the range of the fovea-centered OCT scan may not cover the entire extent of the predominant type of PS (the wide macular type); (3) PS should be differentiated from simple scleral backward bowing, which is commonly observed in highly myopic eyes. The PSH value obtained from the above OCT-based measurements may not represent the actual height of the PS. Thus, we suggest changing the term PSH to macular outward scleral height (MOSH) when applying this quantitative measurement to macular scleral deformation analysis.

In this study, we analyzed 188 highly myopic eyes from 188 participants to investigate the characteristics of MOSH in different grades of MTM, and to explore the risk factors for MTM.

## 2. Patients and Methods

### 2.1. Study Population

In this clinical-based, cross-sectional study, we included participants with high myopia, admitted to the Shanghai General Hospital from January 2015 to December 2018. Approval was obtained from the Ethics Committee of the Shanghai General Hospital to perform this study, and the procedures conformed to the tenets of the Declaration of Helsinki. All the study participants signed informed consent for research purposes prior to the exam.

The inclusion criteria for participant selection were as follows: (1) diagnosis of high myopia, which was defined as an axial length (AL) of ≥26 mm, or a myopic refractive error of ≤−6.0 diopters (D); (2) age >18 years old.

The exclusion criteria were as follows: (1) previous intraocular or refractive surgery, other than cataract surgery; (2) coexisting, or a history of, ocular or severe systemic disease, including dense cataract, glaucoma, diabetic retinopathy, and autoimmune disease; (3) media opacity or poor central fixation leading to poor-quality (image quality <60) OCT images, or the lowest point of the ocular wall not located at the fovea; (4) eyes with dome-shaped macular were excluded, as the protrusion could interfere with the measurement of the macular area.

The grades of MTM referred to the ATN classification system. Based on swept-source optical coherence tomography (SS-OCT), eyes with no macular schisis were defined as the nMTM group. The remaining eyes with MTM were divided into FS, FD, MH, and MHRD groups, respectively. In the MH group, two subgroups were further divided according to the existence of FS around the MH, namely, MH/FS− and MH/FS+ (Figure 1).

### 2.2. SS-OCT Imaging Measurements

The original scanning images of SS-OCT (DRI OCT-1, Topcon Corp, Tokyo, Japan) were obtained using 9 mm radial scans. The central choroidal thickness (CT), scleral thickness (ST), and MOSH were measured using built-in software on SS-OCT, according to the method described previously [7]. The vertical distance from the subfoveal retinal pigment epithelium line to 3 mm nasal, temporal, superior, and inferior was measured using the B-scan OCT image, and was defined as the MOSH (Figure 2).

All the morphologic parameters were measured using DRI OCT Triton software by 2 independent, well-trained observers (J.Y.Y. and H.Z.). In cases of disagreement, an adjudication was made by a retinal specialist (Y.F.). The interobserver intraclass correlation coefficients were excellent for temporal, superior, inferior, and average MOSH (ICC > 0.75, *p* < 0.001), and good for nasal MOSH (ICC = 0.746, *p* < 0.001).

### 2.3. Classification of Atrophic Myopic Maculopathy (AMM)

Fundus photography was obtained from a digital retinal camera built into the same SS-OCT. According to the ATN classification system, AMM was classified into the following five categories: A0—no myopic retinal lesions; A1—tessellated fundus only; A2—diffuse choroidal atrophy; A3—patchy chorioretinal atrophy; A4—complete macular atrophy. A1 and A2 were defined as mild AMM, whereas A3 and A4 were defined as severe AMM.

The classification and grading of AMM were performed by 2 independent, well-trained graders (J.Y.Y. and H.Z.). In cases of disagreement, an adjudication was made by a retinal specialist (Y.F.).

### 2.4. Statistical Analysis

Statistical analysis was performed using SPSS-IBM version 22.0 software (IBM Corp., Armonk, NY, USA). The participants’ characteristics are presented as counts or proportions for categorical data, and as means ± standard deviation for continuous data. The frequency of sex was compared using Pearson’s χ^2^ test. The Wilcoxon rank sum test, or one-way analysis of variance, was used to detect differences in age, refractive error, visual acuity, AL, CT, ST, and MOSH among MTM groups; for comparison between different categories of AMM, the Kruskal–Wallis rank test was used. The independent samples t-test, or the Mann–Whitney U test, was used to compare the MTM group and the nMTM group, and to compare the two subgroups of MH. Logistic regression models and the area under the ROC curve (AUC) were calculated to determine the potential risk factor for MTM. All *p* values were 2-sided, and a *p* value of <0.05 was considered statistically significant.

## 3. Results

### 3.1. Clinical Characteristics

A total of 188 eyes from 188 participants with high myopia were examined in this study. Table 1 presents the general characteristics of all the participants. The average age (*p* = 0.503), gender distribution (*p* = 0.338), and SER (*p* = 0.338) did not differ among the five groups. However, significance was found in the AL (*p* = 0.046) and BCVA (*p* < 0.001). Specifically, the AL in the MHRD group was significantly higher than in the other groups. The nMTM and FS groups showed the highest BCVA, followed by the MH and FD groups. The MHRD group displayed the poorest BCVA. Moreover, for eyes compared between the nMTM and MTM groups, BCVA in the MTM group was predominantly worse than in the nMTM group, while no significant difference was found in AL (*p* = 0.295) (Table 1).

### 3.2. Morphological Characteristics Defined by SS-OCT

The morphological characteristics measured through SS-OCT, such as CT, ST, and MOSH, for different grades of MTM groups are shown in Table 2. Compared with the nMTM group, the MTM groups displayed a significantly thinner CT (*p* < 0.001), a significantly thicker ST (*p* = 0.013), and a significantly higher MOSH (*p* < 0.001) in all four quadrants. Interestingly, we found that CT and ST did not differ between the nMTM and MH groups. Additionally, CT (*p* < 0.001) and ST (*p* < 0.001) were significantly higher in the nMTM and MH groups than in the FS, FD, and MHRD groups. MOSH was found to be significantly lower in the nMTM group than in the FS, FD, and MHRD groups in all four quadrants (*p* < 0.001); however, no significant difference was found in MOSH among the FS, FD, and MHRD groups. Moreover, MOSH in the MH group showed a similar tendency as in the nMTM group (Table 2). The maximum MOSH was found to be significantly lower in the nMTM group than in the MTM groups (*p* < 0.001). Figure 3 shows the distributions of the maximum MOSH for the five study groups. The most common location of the maximum MOSH was in the temporal quadrant in all groups. The sum of the proportion of eyes with the maximum MOSH, located on the nasal and temporal side in the MTM group (79.17%), was significantly higher than in the nMTM group (56.67%) (*p* = 0.009).

Considering the similarities between OCT morphological characteristics in the MH and nMTM groups, we separated the MH eyes from the other MTM groups for analysis. The MH group was further divided into the macular hole with foveoschisis (MH/FS+) subgroup and the macular hole without foveoschisis (MH/FS−) subgroup. ST was found to be significantly thicker in the MH/FS− subgroup than in the MH/FS+ subgroup (*p* = 0.003). MOSH was found to be significantly lower in the MH/FS− subgroup than in the MH/FS+ subgroup, in the nasal (*p* = 0.008) and inferior (*p* = 0.026) quadrants, and the average MOSH was also lower in the MH/FS− subgroup (*p* = 0.012). Moreover, similar general and morphological characteristics were found between the MH/FS− subgroup and the nMTM group (Table 3). 

### 3.3. Atrophic Myopic Maculopathy

The constituent ratios for AMM differed significantly between the MTM and nMTM groups (*p* = 0.020). The proportion of eyes with A3 or A4 in the MTM group (28.48%) was significantly higher (*p* = 0.003) than in the nMTM group (3.33%). Figure 4 shows the distributions of AMM grades for the five groups. Eyes with A1 accounted for 16.67% of all eyes in the MH group; this proportion was similar to that in the nMTM group (16.67%), but significantly higher than that in the FS (5.88%) and FD (6.82%) groups (*p* = 0.029). No eyes with A1 were observed in the MHRD group. The prevalence of severe AMM (A3 or A4) was the highest in the MHRD group (47.37%), followed by the FD group (29.54%), MH group (19.04%), and FS group (17.64%) (*p* < 0.001). Compared with the MH/FS− subgroup, eyes in the MH/FS+ subgroup suffered more severe AMM (*p* = 0.009).

### 3.4. Risk Factors for MTM

A binary logistic regression model was used to analyze the risk factors for MTM. The eyes were classified into two groups, based on the presence or absence of MTM. The FS, FD, and MHRD groups, and the MH/FS+ subgroup, were combined to form the MTM group, owing to similarities in their general and morphological characteristics. Comparisons between this newly constructed group and the nMTM group showed that MOSH is an independent risk factor for MTM (OR = 1.108, 95% CI: 1.058–1.162, *p* < 0.001). Furthermore, the largest AUC for MTM occurrence was observed for the average MOSH (0.857), followed by the nasal MOSH (0.832), temporal MOSH (0.799), superior MOSH (0.797), and inferior MOSH (0.794). An average MOSH of >513.5 μm predicted MTM onset with a sensitivity of 78.8% and a specificity of 80.8% (Figure 5). To explore the risk factors of severe MTM, the eyes were segregated by the presence or absence of RD. The logistic regression analysis revealed that MOSH (OR = 1.031, 95% CI: 1.008–1.054, *p* = 0.007) is an independent risk factor of RD.

## 4. Discussion

This study first reported that an OCT-based measurement of MOSH was a risk factor for the occurrence of MTM, and also a risk factor for severe MTM. Posterior staphyloma (PS) has been recognized as an important risk factor for severe alterations and a high prevalence of MTM [9,10,11]. However, standardized methods to precisely measure the curvature of PS have not been developed. Three-dimensional magnetic resonance imaging has been used to quantify the morphology of the ocular contour [12], but it cannot perform quantitative assessments of PS. In this study, we used SS-OCT to quantitatively evaluate the macular scleral morphology in different MTM types, by measuring MOSH in all four quadrants of the 6 mm macular area (Figure 2).

Our results showed significant differences in MOSH between the MTM and nMTM groups in several quadrants; however, no significant differences in AL were detected between the groups (Table 2). The MTM eyes appeared to have a steeper curvature of posterior scleral shape. We deduced that the curvature change, especially in the macula, may play a more important role than the elongation of AL in MTM formation. Previous studies have reported that retinoschisis is usually confined to the area of PS, and they have inferred that the outer layer of retinoschisis is related to the vertical traction of PS [10,13], which supports our hypothesis. Moreover, the degree of posterior scleral deformation was not uniform throughout the posterior pole. Our results show that the maximum MOSH tended to be located in the temporal or nasal quadrants, rather than in the superior or inferior quadrants, in MTM eyes, which indicates that the deformation of MTM eyes commonly expanded in the horizontal direction, not just by simple elongation. The significantly asymmetric MOSH around the macula may lead to MTM progression. CT and ST in this study were found to be significantly thinner in the MTM group than in the nMTM group, which is comparable to a recent report [14]. One possible mechanism is the passive reorganization of the intrinsic extracellular matrix, collagen bundles, and scleral lamellae while the eye enlarges. In addition, the stretching of the Bruch’s membrane may be the driving force for the thin CT.

Among the MTM groups, ST and MOSH in the MH group were found to differ significantly from those in the other MTM groups. Jo et al. proposed that myopic MH could be classified into two categories, namely, those “with macular schisis” and those “without macular schisis” [15]. MH with retinoschisis is usually observed in eyes with severe PS, and is associated with a low postoperative closure rate and a high risk of RD. Recently, a study detected differences in the SER, AL, and frequency of staphyloma between MH with and without macular schisis, which suggests a potential difference in the pathogenesis of these two subtypes [16]. In the MH group, we observed that the MH with FS had a steeper scleral curvature, similar to the degree of scleral curvature in the FD and MHRD groups (Table 3). However, the MH without FS displayed a relatively flat scleral curvature, which was similar to that in the nMTM group. These differences may be attributable to the various directions of mechanical traction. An eye classified as MH/FS− is primarily pulled by the epiretinal membrane in a tangential direction, whereas an eye classified as MH/FS+ is subjected to dual effects, including vertical PS dilatation and vitreomacular traction. Therefore, MH with FS is more likely to develop into MHRD during disease progression. A study showed that myopic MH without FS and idiopathic MH display similar, high postoperative closure rates [17]. Therefore, it may be more appropriate to define the MH group in MTM as myopic MH with FS, while classifying myopic MH without FS as idiopathic MH.

AMM is an important characteristic in the progression of high myopia. Chorioretinal atrophy in the macular area may be related to the occurrence of FS and FD in high myopia [18]. A recent study found that the incidence of FS in A2 myopic maculopathy is significantly higher than in A1 myopic maculopathy [19]. Our previous study identified different grades of AMM in various MTM categories [20,21]. In the present study, the MTM groups suffered significantly more severe AMM than the nMTM group. The proportion of severe AMM (A3 and A4) significantly increased as the MTM became worse (Figure 4). It is suggested that the attenuation of choroid in the severe MTM group may lead to insufficient blood perfusion and impaired Müller cell function, which aggravates foveal lesions and atrophic maculopathy. This result may also explain the rarity of a severe grade of MTM in young participants with high myopia, whereas the incidence of FD or MHRD is high in older participants, because AMM progresses with continued AL elongation. One possible hypothesis for this observation is that the RPE structure and barrier system function are damaged due to the Burch’s membrane defect in severe AMM, which may cause the accumulation of subretinal fluid and the progression of mild MTM into severe lesions [22].

The natural progression of MTM is believed to begin with FS, then develop into FD or full-thickness MH, and finally progress to RD. However, the risk factors for the onset and progression of MTM have not been well studied. Using the binary logistic regression analysis, we found that MOSH was associated with the formation of MTM, and was also identified as an independent risk factor for severe MTM (Figure 5). When PS deepens and widens, the prevalence of chorioretinal atrophy may increase [23]. AMM and MOSH may interact, and, together, affect the progression of MTM. Both OCT and fundus photography may be required in routine follow-ups to monitor the deformation of the posterior pole of the eyeball and the progression of AMM. In this case, the occurrence and progression of MTM may be monitored in participants with high myopia, and timely clinical interventions can be conducted to prevent severe visual impairment.

The current study has some potential limitations. First, this was a case–control study with relatively small sample sizes; thus, long-term studies with large sample sizes are required to further observe the development of MTM. Second, the hospital-based highly myopic participants in the study might have caused selection bias. Finally, we did not analyze the condition of the vitreomacular interface, which may have influenced MTM progression, especially in regards to paravascular vitreal adhesion. Wide-field OCT, covering both deeper and wider scan areas, has the advantage of detecting postequatorial adhesions, which could be used in further research.

## 5. Conclusions

In summary, to the best of our knowledge, this study was the first to quantitatively evaluate MOSH in eyes with different MTM categories. We found that MOSH would be more suitable for predicting MTM than axial length. An OCT-based MOSH > 513.5 μm is a risk factor for the occurrence of MTM. As severe AMM was associated with the progression of MTM, it is important to evaluate baseline AMM in the clinical management of MTM. The OCT morphological characteristics of the macular hole without foveoschisis group were different from the other MTM groups, suggesting a distinct pathology from macular hole with foveoschisis.

## Figures and Tables

**Figure 1 jcm-11-01599-f001:**
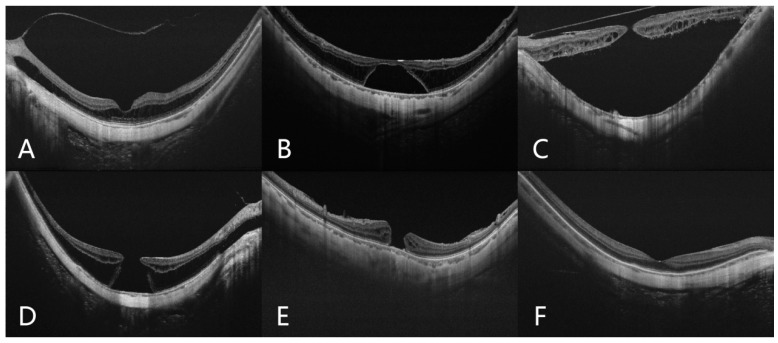
Typical SS-OCT images of highly myopic eyes with different types of MTM. (**A**) Foveoschisis; (**B**) foveal detachment; (**C**) macular hole retinal detachment; (**D**) macular hole with foveoschisis; (**E**) macular hole without foveoschisis; (**F**) no MTM.

**Figure 2 jcm-11-01599-f002:**
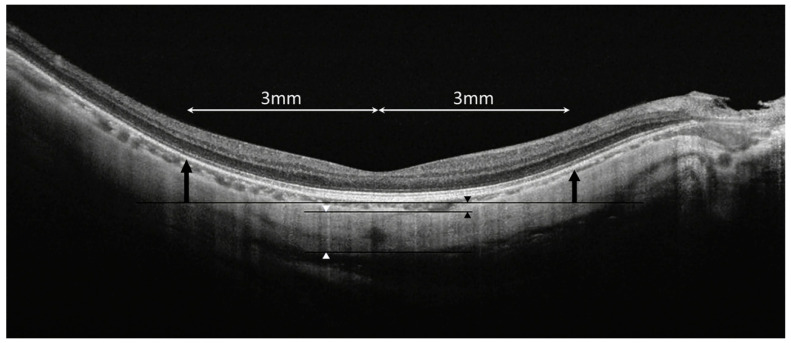
Diagram of SS-OCT image measurement. The macular outward scleral height (black arrows), central choroidal thickness (black arrowheads), and central scleral thickness (white arrowheads) were measured using DRI OCT Triton software built in SS-OCT.

**Figure 3 jcm-11-01599-f003:**
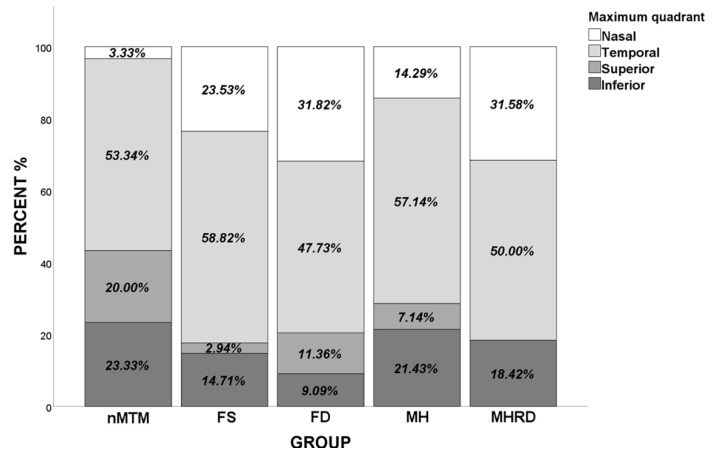
Distribution of the maximum MOSH, located in different quadrants in different grades of MTM. MTM: myopic tractional maculopathy, FS: foveoschisis, FD: foveal detachment, MH: macular hole, MHRD: macular hole with retinal detachment.

**Figure 4 jcm-11-01599-f004:**
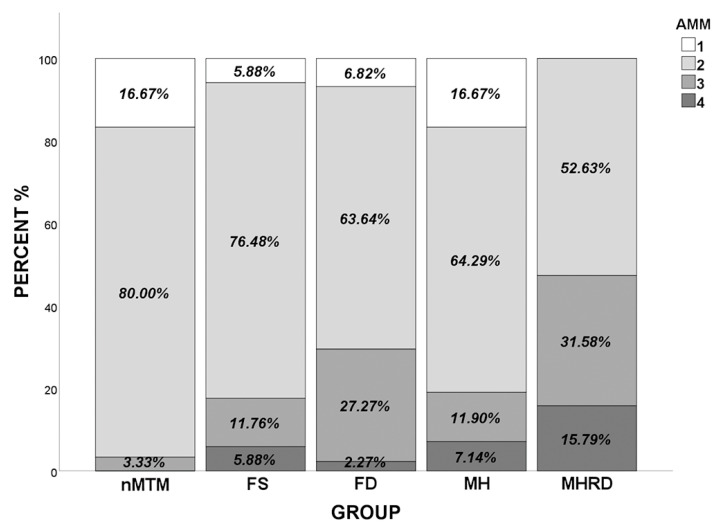
Distribution of the different categories of atrophic myopic maculopathy in four different types of MTM and in highly myopic eyes without MTM. AMM: atrophic myopic maculopathy, MTM: myopic tractional maculopathy, FS: foveoschisis, FD: foveal detachment, MH: macular hole, MHRD: macular hole with retinal detachment.

**Figure 5 jcm-11-01599-f005:**
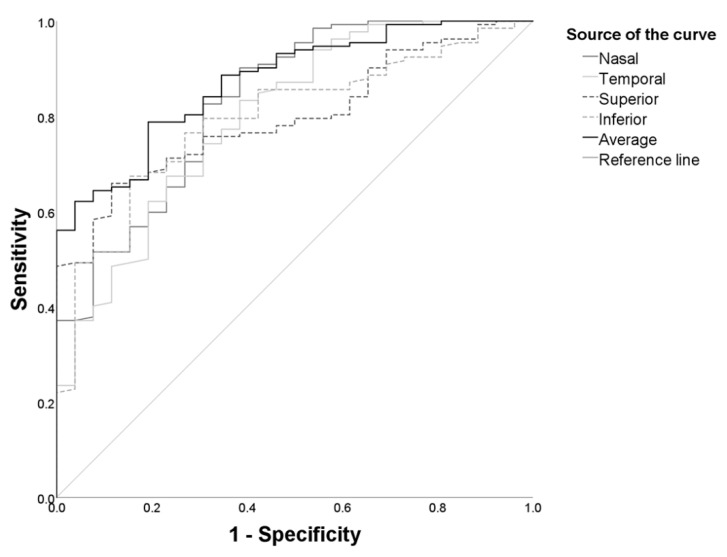
ROC analysis of different MOSH parameters. The average MOSH showed the highest AUC value. AUC: area under the curve, MOSH: macular outward scleral height, ROC: receiver operator characteristic.

**Table 1 jcm-11-01599-t001:** General characteristics among highly myopic eyes with different grades of MTM.

	nMTM	MTM				*p* *	*p* ^†^	Post hoc ^‡^
		FS	FD	MH	MHRD			
No. of eyes	30	34	44	42	38			
Male, *n* (%)	7	7	12	4	8	0.338	0.642	
Age, y	58.7 ± 9.3	60.6 ± 8.1	62.0 ± 10.4	62.4 ± 9.1	61.0 ± 9.0	0.503	0.120	
AL, mm	29.22 ± 1.30	29.30 ± 1.42	29.68 ± 1.42	29.16 ± 2.01	30.34 ± 1.84	0.046	0.295	nMTM, FS, MH < MHRD
SER, D	−13.1 ± 4.1	−12.5 ± 5.0	−13.8 ± 3.8	−13.3 ± 3.5	−15.7 ± 4.4	0.338	0.695	
BCVA, logMAR	0.66 ± 0.42	0.55 ± 0.45	1.32 ± 0.63	1.25 ± 0.42	1.69 ± 0.63	<0.001	<0.001	FS, nMTM < FD, MH < MHRD

MTM: myopic tractional maculopathy, FS: foveoschisis, FD: foveal detachment, MH: macular hole, MHRD: macular hole with retinal detachment, AL: axial length, SER: spherical equivalent refraction, D: diopter, BCVA: best-corrected visual acuity, logMAR: logarithm of minimal angle of resolution. * *p* value for the difference among the five groups based on one-way ANOVA, Kruskal–Wallis rank tests, or χ^2^ tests, as appropriate. ^†^
*p* value for the difference between the nMTM and MTM groups, based on Student’s *t*-tests. ^‡^ Multiple comparisons among the five groups.

**Table 2 jcm-11-01599-t002:** Morphologic parameters of SS-OCT among highly myopic eyes with different grades of MTM.

	nMTM	MTM				*p* *	*p* ^†^	Post hoc ^‡^
		FS	FD	MH	MHRD			
No. of eyes	30	34	44	42	38			
CT, μm	50.8 ± 25.6	28.1 ± 13.3	23.8 ± 20.3	42.1 ± 42.8	23.1 ± 17.6	<0.001	<0.001	FS, FD, MHRD < nMTM, MH
ST, μm	283.7 ± 84.3	227.2 ± 59.9	250.1 ± 71.4	279.0 ± 93.5	213.7 ± 73.5	<0.001	0.013	FS, FD, MHRD < nMTM, MH
MOSH, μm								
nasal	316 ± 200	635 ± 210	615 ± 229	473 ± 269	663 ± 293	<0.001	<0.001	nMTM < MH < FS, FD, MHRD
temporal	478 ± 210	743 ± 202	712 ± 207	601 ± 222	777 ± 169	<0.001	<0.001	nMTM < MH < FS, FD, MHRD
superior	375 ± 147	587 ± 174	573 ± 197	452 ± 191	554 ± 186	<0.001	<0.001	nMTM < MH < FS, FD, MHRD
inferior	416 ± 136	565 ± 192	612 ± 206	519 ± 214	678 ± 216	<0.001	<0.001	nMTM < MH, FS < FD, MHRD
average	399 ± 137	632 ± 144	628 ± 155	509 ± 192	665 ± 156	<0.001	<0.001	nMTM < MH < FS, FD, MHRD
maximum	535 ± 173	800 ± 204	801 ± 209	662 ± 228	874 ± 209	<0.001	<0.001	nMTM < MH < FS, FD, MHRD

MTM: myopic tractional maculopathy, FS: foveoschisis, FD: foveal detachment, MH: macular hole, MHRD: macular hole with retinal detachment, CT: choroid thickness, ST: sclera thickness, MOSH: macular outward scleral height. * *p* value for the difference among the five groups, based on one-way ANOVA or Kruskal–Wallis rank tests, as appropriate. ^†^
*p* value for the difference between the nMTM and MTM groups, based on Student’s *t*-tests. ^‡^ Multiple comparisons among the five groups.

**Table 3 jcm-11-01599-t003:** Comparisons among the nMTM group and the MH/FS− and MH/FS+ subgroups in highly myopic eyes.

	nMTM	MH		*p* *	*p* ^†^
		MH/FS−	MH/FS+		
No. of eyes	30	15	27		
Age, y	58.7 ± 9.3	60.1 ± 8.3	63.6 ± 9.4	0.616	0.242
AL, mm	29.22 ± 1.30	28.97 ± 2.17	29.28 ± 1.94	0.710	0.668
SER, D	−13.1 ± 4.1	−13.6 ± 4.2	−13.2 ± 3.4	0.817	0.812
CT, μm	50.8 ± 25.6	56.0 ± 62.3	34.4 ± 25.1	0.760	0.216
ST, μm	283.7 ± 84.3	350.8 ± 102.8	241.7 ± 62.8	0.030	0.003
MOSH, μm					
nasal	316 ± 200	319 ± 270	560 ± 232	0.961	0.008
temporal	478 ± 210	513 ± 229	650 ± 206	0.613	0.055
superior	375 ± 147	368 ± 193	487 ± 187	0.775	0.114
inferior	416 ± 136	422 ± 218	573 ± 195	0.912	0.026
average	399 ± 137	411 ± 195	563 ± 170	0.808	0.012

MTM: myopic tractional maculopathy, FS: foveoschisis, MH: macular hole, AL: axial length, SER: spherical equivalent refraction, D: diopter, CT: choroid thickness, ST: sclera thickness, MOSH: macular outward scleral height. * *p* value for the difference between the nMTM group and the MH/FS− subgroup, based on Student’s *t*-tests. ^†^
*p* value for the difference between the MH/FS− and MH/FS+ subgroups, based on Student’s *t*-tests.

## Data Availability

The presented data in this study are available from the corresponding author upon reasonable request.

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
