# Peer review of "Association between Different Grades of Myopic Tractional Maculopathy and OCT-Based Macular Scleral Deformation"

_jcm, 2022, doi:10.3390/jcm11061599_

Round 1

Reviewer 1 Report

Authors investigated that macular outward scleral height (MOSH)  would be more suitable for estimating myopic tractional maculopathy (MTM) occurrence than axial length. Grading of atrophic myopic maculopathy (AMM) helps evaluate the severity of MTM. The categorization of The macular hole without foveoschisis (MH/FS-) as a distinct grade from MH/FS+ might be preferable. It is reasonable and the manuscript is written well. The measurement method is correct. I think this conclusion is similar to the paper https://doi.org/10.1038/s41598-018-22759-y. This paper is minor to the Ohno faction, but is very similar to the results of this current study.

Minor indication.

Line 25, the two lines above say "P = 0.003", but why is it written as "P <0.05"?
Line 83, what are the manufacturer and model name of the SS-OCT?

Reviewer 2 Report

I would like to congratulate the authors for undertaking this very interesting study. 

In this paper, the Authors used the strategy characteristics of macular outward scleral height (MOSH) in different grades of myopic tractional maculopathy (MTM), with such strategies, risk factors for MTM can be found. 

This sounds very interesting, as the target parameter for the physician is not just the number  but the thanks to the stratification into different appearance of the macula can be applied in real life. 

The paper is well written and concise; however, I have some remarks which should be addressed in a revised version:

The table 1 look really interesting.  However it is quite messy and hard to read. especially the age value is impossible to determine. This also applies to table 2. 

1st my main comment is that you should discuss more your observation of thickness measurements in studied groups , add some animal studies (if they exist), hypothesize more about your observation.

Why was the (Image Quality set as<60 usually the borderline is drafted below 70? How many eyes were not included due to the low QI?

The main message of this paper is that analysis of wide-field SS-OCT of MOHS is suitable for estimating MTM.
